# Video-SVD: Efficient Video Diffusion via Orthogonal Basis Composition

**Zhang Wan** [1 2]  **Yu Li** [1 2]  **Tianze Huang** [1 2]  **Haochen Li** [1 2]  **Juan Cao** [1 2]  **Sheng Tang** [1 2]

## Abstract

Video Diffusion Transformers (VDiTs) represent the state of the art in video generation but remain constrained by the quadratic complexity of dense self-attention. To address this attention bottleneck, we analyze the pre-softmax matrix ($QK^\top$) and reveal two key properties: (1) video attention exhibits an effective low-dimensional structure with rapid singular-value decay, and (2) real motion induces hybrid spatio-temporal patterns rather than rigid "spatial vs. temporal" layouts. Guided by these observations, we propose Video-SVD, a training-free and plug-and-play acceleration method that does not modify the original network parameters. Video-SVD learns checkpoint-adaptive orthogonal bases offline and, at inference time, replaces expensive dense attention computation with lightweight online subspace projection and basis composition. To preserve high fidelity, Video-SVD further employs layer-shared dual-stream residual modules to recover fine-grained content details and positional information. Across HunyuanVideo and Wan2.1 backbones, Video-SVD achieves significant end-to-end speedups while maintaining high visual quality, reaching 1.92× on HunyuanVideo, 1.75× on Wan2.1-1.3B, and 1.79× on Wan2.1-14B.

## 1. Introduction

Video Diffusion Transformers (VDiTs) (Peebles & Xie, 2023) have defined the new state-of-the-art in generative AI, producing high-fidelity and temporally coherent video content, as evidenced by models like Wan 2.1 (Wan et al., 2025), HunyuanVideo (Kong et al., 2024), and Cogvideo (Hong et al., 2023). However, this success rests on a fundamentally constrained foundation: the quadratic computational complexity of the self-attention mechanism. As video resolution

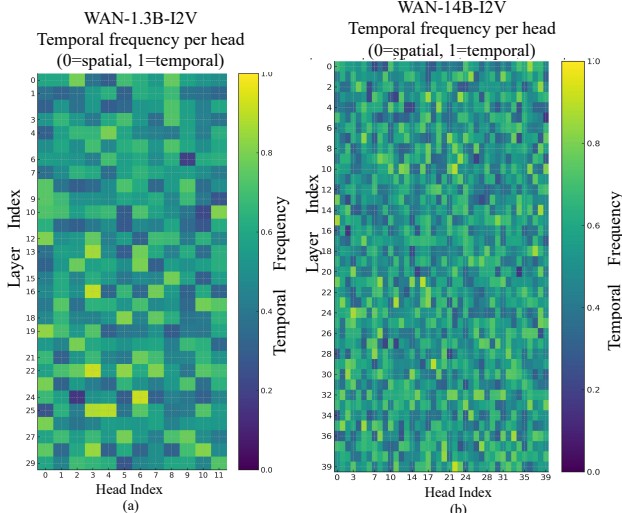

*Figure 1.* **Head-wise temporal frequency heatmaps for open-source video generators.** Values range from 0 (purely spatial) to 1 (purely temporal): (a) WAN-2.1-1.3B, (b) WAN-2.1-14B. Our analysis reveals that approximately 71.4% of all heads reside within the $[0.3, 0.7]$ "mixed band," indicating that the vast majority of attention layers favor pervasive spatio-temporal interactions over a rigid spatial/temporal dichotomy.

and duration increase, the sequence of spatio-temporal tokens $n$ grows explosively (e.g., $n$ can exceed 20,000 for a 480p, 81-frame video), making this quadratic computation the primary bottleneck for efficient, high-resolution, and long-video synthesis. For instance, generating a 5-second video on a top-tier H100 GPU can take nearly 1000 seconds (Sun et al., 2025), severely limiting practical deployment.

To address this limitation, researchers have investigated several routes for acceleration. One set of techniques (Song et al., 2023; Wang et al., 2024a; Yin et al., 2024) targets the expensive cost of sampling, utilizing the self-consistency characteristic of the probability flow ODE (PF-ODE) (Song et al., 2021) for distillation, which substantially reduces the necessary sampling iterations (Lv et al., 2025). Different studies (Liu et al., 2025; Zhao et al., 2025) employ strategies for caching intermediate features, a technique that enhances sampling velocity requiring no additional training. A separate, vital direction seeks to lessen the computational burden of the self-attention mechanism. Fundamentally, the Dense Interaction Matrix (derived from $QK^T$) quantifies the relationship between arbitrary spatio-temporal tokens, determining how semantic information flows across the video.

[1]Institute of Computing Technology, Chinese Academy of Sciences, Beijing, China [2]University of Chinese Academy of Sciences, Beijing, China. Correspondence to: Yu Li <liyu@ict.ac.cn>.

*Proceedings of the 43rd International Conference on Machine Learning*, Seoul, South Korea. PMLR 306, 2026. Copyright 2026 by the author(s).

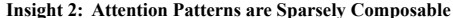

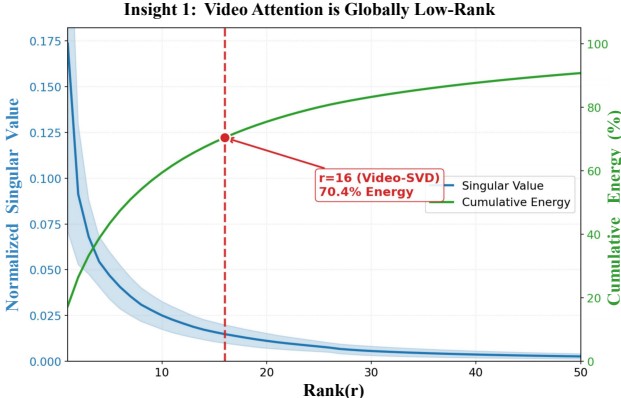

*Figure 2.* **Video attention ($QK^T$) exhibits effective low-dimensional structure.** We perform Singular Value Decomposition (SVD) on $QK^T$ matrices to analyze their underlying structure. The blue curve shows a steep, rapid decay of normalized singular values on a linear scale, illustrating heavy information redundancy. The green curve (right axis) represents the cumulative energy ratio, where the first $r = 16$ singular values (red dashed line) are sufficient to capture 70.4% of the total energy. This indicates that the attention matrix is highly compressible, supporting the motivation for our Video-SVD approach.

To reduce the quadratic complexity of modeling these dense interactions, existing VDiT acceleration methods (Beltagy et al., 2020; Sun et al., 2025; Xi et al., 2025) primarily rely on sparse priors.

State-of-the-art approaches (e.g., SVG (Xi et al., 2025), VORTA (Sun et al., 2025)) observe that attention patterns often exhibit two distinct layouts: **Spatial Heads**, which show block-wise patterns by focusing on tokens within the same frame, and **Temporal Heads**, which show slash-wise patterns by attending to tokens at the *same spatial coordinates* across different frames. Based on this, they adopt a Rigid Classification paradigm: hard-assigning each head to be either "Spatial" or "Temporal" and pruning tokens outside these predefined scopes.

While this strategy effectively reduces redundancy in static or slow-motion scenes, we argue that it is suboptimal for modeling complex dynamics. In high-fidelity generation, real motion implies that objects shift spatial coordinates over time (e.g., a car moving from left to right), creating intricate spatio-temporal entanglements. This forms a "Grey Zone"—complex hybrid interactions that are neither purely spatial nor purely location-fixed temporal. As shown in Fig. 1, the prevalent attention heads in the 0.3-0.7 frequency band represent this essence of motion—hybrid spatio-temporal trajectories. Pigeonholing these fluid patterns into rigid binary categories leads to severe information loss and temporal incoherence.

To address this, we move beyond the sparsity assumption. Unlike pruning methods that risk information loss by discarding tokens, we seek to preserve full context while reducing computational redundancy. We propose a fundamental

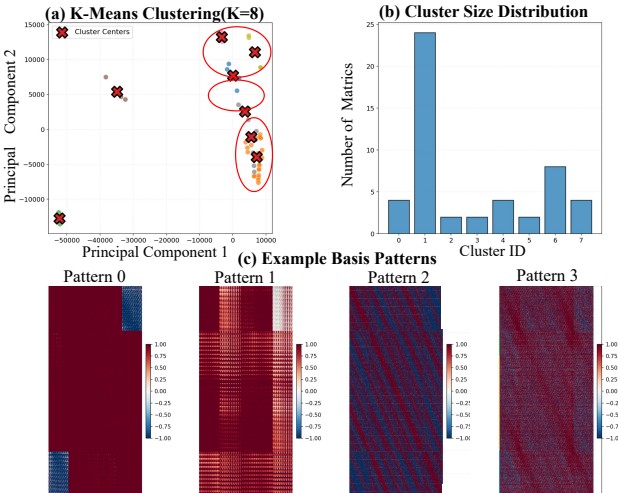

*Figure 3.* **Insight 2: Attention Patterns are Sparsely Composable.** (a) K-Means clustering on $QK^T$ matrices confirms that patterns group into a small set of archetypes (red circles). The density of attention heads within the transition zones between these archetypes shows that attention matrices are continuous linear combinations of hybrid bases rather than binary choices. (b) The cluster distribution is highly non-uniform. (c) Cluster centroids form a shared dictionary capturing semantically meaningful basis patterns: Global Context (Basis 0), Spatiotemporal Grids (Basis 1), and Motion Trajectories (Basis 2 & 3).

inquiry: **Are the dense interactions between all tokens strictly independent, or do they share an intrinsic low-dimensional structure?** This first-principles analysis led to two key discoveries:

- **Insight 1: Video Attention Exhibits Effective Low-Dimensional Structure.** First, our SVD analysis of a large-scale collection of $QK^T$ matrices (Fig. 2) reveals that singular values exhibit a rapid initial decay. This energy concentration indicates that dense attention patterns contain substantial redundancy and can be captured by a compact set of dominant components. In video, many tokens correspond to a small number of shared semantic entities, such as background regions, object parts, and motion trajectories, making many rows and columns of the pre-softmax interaction matrix highly correlated. The top-$K$ components therefore capture the global semantic skeleton, while fine-grained content details and positional information remain as residual components to be recovered.

- **Insight 2: Checkpoint-Adaptive Bases Capture Complex Hybrid Interactions.** Second, we investigate the topology of learned bases to extend the "Spatial vs. Temporal" dichotomy used in prior works. While rigid classification effectively captures static or slow-motion patterns, our learned bases (Fig. 3(c)) reveal that real-world dynamics are far more nuanced. Visualizing these bases exposes distinct Activation (Red)

and Suppression (Blue) zones that form intricate hybrid structures: Global Context filters (e.g., Basis 0) manifest as large-scale activation blocks; Spatiotemporal Grids (e.g., Basis 1) appear as interlaced vertical and horizontal bands; and Motion Trajectories (e.g., Basis 2 & 3) emerge as continuous red diagonal stripes capturing cross-frame object displacement. The cluster distribution (highlighted by red circles in Fig. 3(a)) further confirms that real attention heads reside in the transition zones between these archetypes, showing that attention matrices are continuous linear combinations of shared structural motifs rather than a binary choice.

Motivated by these insights, we propose Video-SVD. Unlike classification-based methods, Video-SVD replaces expensive dense interaction computation with efficient basis composition via a three-stage architecture:

1. **[Offline] Checkpoint-Adaptive Basis Learning:** We extract orthogonal bases from large-scale video data for each target checkpoint and resolution bucket. These bases capture shared low-dimensional attention motifs, serving as a reusable structure for efficient reconstruction after one-time calibration.

2. **[Online] Dynamic Weight Acquisition:** At inference, we avoid constructing the full interaction matrix by dynamically solving for basis weights from sparsely sampled token interactions. This online projection estimates how the current attention head composes the learned bases, substantially reducing the cost of dense $QK^T$ computation.

3. **[Online] High-Frequency Compensation:** To bridge the gap between low-dimensional approximation and full-fidelity attention, we employ lightweight networks to dynamically synthesize the missing residuals, ensuring precise restoration of fine-grained content details and positional information.

Video-SVD is designed for method-level generalization across VDiT backbones: the same pipeline of offline basis extraction, online projection, and residual compensation applies to different models, while the basis bank is calibrated once for each target checkpoint and resolution bucket. By replacing dense matrix multiplication with efficient basis composition and structured residual compensation, Video-SVD achieves significant acceleration while maintaining high fidelity even in complex video dynamics.

Our contributions are as follows:

- We uncover the intrinsic low-dimensional structure of VDiT attention and identify checkpoint-adaptive

bases of hybrid patterns, enabling precise modeling of complex motion.

- We propose Video-SVD, reducing attention complexity via offline basis learning, online projection, and residual compensation.

- Video-SVD achieves $1.92\times$ and $1.79\times$ end-to-end acceleration on HunyuanVideo and Wan 2.1-14B, respectively, while maintaining a PSNR of 29.5 and 28.0, outperforming state-of-the-art methods.

## 2. Related Work

**Step Reduction, Caching, and Token Merging**    To accelerate the denoising process, advanced solvers and distillation techniques (Wang et al., 2024b) minimize the required sampling steps. Exploiting temporal redundancy, methods like DeepCache (Ma et al., 2024) and Faster Diffusion (Li et al., 2024) cache intermediate features to skip high-level computations, while TGATE (Zhang et al., 2024) caches attention outcomes. Additionally, ToMe (Bolya et al., 2023) reduces sequence length by merging similar tokens. Our Video-SVD is fully compatible with these approaches, enabling combined multiplicative speedups.

**Sparse Attention**    Optimizing the quadratic attention bottleneck is critical for high-resolution generation. In the video domain, methods like VORTA (Sun et al., 2025), SVG (Xi et al., 2025), SLA (Zhang et al., 2025a), SVG2 (Yang et al., 2025), and Radial Attention (Li et al., 2025) reduce attention cost through sparsity or token interaction priors. VORTA and SVG adopt a *Rigid Classification* paradigm that assigns attention heads to predefined spatial or temporal buckets, while Radial Attention introduces a geometric locality prior and SVG2 improves sparse execution through semantic token organization. However, these approaches depend on handcrafted sparse patterns and often customized CUDA kernels, which can oversimplify hybrid motion and increase reproduction difficulty. In contrast, Video-SVD reconstructs dense attention through data-driven basis composition and residual compensation, preserving global context while reducing redundant computation.

**Linear and Low-Rank Approximations**    Linear Attention (Shen et al., 2021; Xie et al., 2024), Linformer (Wang et al., 2020), and related low-rank approximation methods reduce attention complexity via kernel approximations, static projections, or landmark-based reconstruction. These methods provide important evidence that attention contains compressible structure, but generic approximations often struggle to preserve high-fidelity spatio-temporal details in video diffusion. In contrast, Video-SVD makes low-dimensional attention reconstruction practical for video

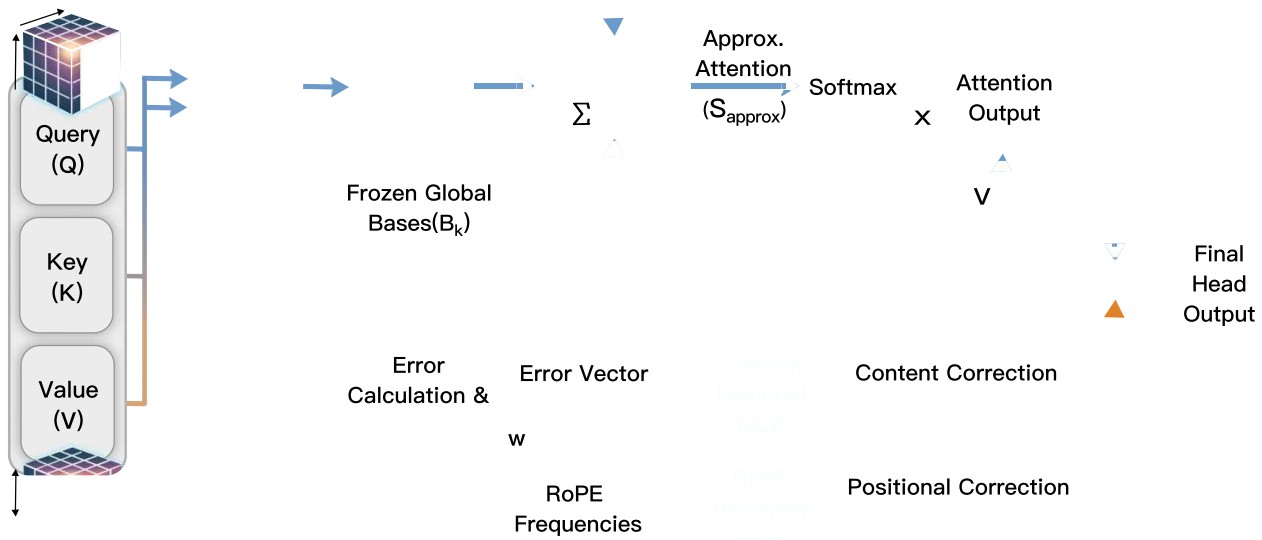

*Figure 4.* **Architecture of Video-SVD.** Our method replaces the $O(n^2d)$ global attention with a three-stage pipeline: in Online Reconstruction (Stages 1 & 2), the core path approximates the attention matrix $S_{\text{approx}}$ by linearly composing checkpoint-adaptive bases; in Dual-Stream Residual Compensation (Stage 3), parallel MLPs employ error vectors and RoPE frequencies to rectify content errors and restore positional information; the final head output integrates the reconstructed path with these high-fidelity residual components.

generation by combining checkpoint-adaptive offline basis learning, online coefficient estimation, and structured residual compensation. Rather than relying on a fixed projection or kernel approximation, Video-SVD learns data-driven bases that capture shared video attention motifs and uses lightweight residual modules to restore fine-grained content and positional information.

## 3. Methodology

To fundamentally overcome the $O(N^2d)$ computational bottleneck of the self-attention mechanism in Video Diffusion Transformers (VDiTs), we propose Video-SVD, a dynamic attention decomposition framework. Guided by the insight that video attention matrices exhibit effective low-dimensional structure, this framework reformulates the expensive dense matrix computation into a three-stage paradigm: "Offline Checkpoint-Adaptive Basis Learning + Online Sampling Projection + Structured Error-Aware Compensation."

### 3.1. Stage 1: Offline Checkpoint-Adaptive Basis Learning

Our core hypothesis is that complex spatio-temporal attention patterns can be represented as linear combinations of a compact set of prototype orthogonal bases. For each target checkpoint and resolution bucket, Video-SVD performs a one-time calibration to extract a reusable basis bank, which is then shared across subsequent inference.

**Streaming Incremental Decomposition.** We construct a large-scale dataset of attention matrices sampled from pretrained VDiT models. Given the high dimensionality of full matrices (e.g., $N > 20,000$ due to long temporal sequences), loading all matrices into memory is computationally infeasible. Therefore, we employ an **Incremental SVD** algorithm to process flattened matrices in a streaming fashion. This technique iteratively updates the singular subspace using mini-batches of incoming data, allowing us to extract checkpoint-adaptive orthogonal bases $\{B_k\}_{k=1}^{K}$ with limited memory. These bases capture shared low-dimensional structural motifs of video attention and possess strict mathematical orthogonality ($\langle B_i, B_j \rangle \approx \delta_{ij}$).

**Implementation Strategy** To ensure production-grade fidelity, we align our strategy with industrial protocols:

- **Resolution Bucketing:** Since attention patterns are sensitive to sequence length $N$, we learn a distinct set of bases for each standard resolution bucket (e.g., 720p) to avoid interpolation artifacts.

- **Strided Basis Sharing:** Observing high **structural similarity** between adjacent layers, we share one set of bases across groups of adjacent layers, significantly

reducing basis storage without performance degradation.

### 3.2. Stage 2: Online Dynamic Weight Calculation

During inference, to circumvent the computationally prohibitive $O(N^2 d)$ overhead of calculating the full $QK^T$ matrix, we propose a sampling-based subspace projection strategy. This approach resolves the basis weights $\mathbf{w}$ for the current attention head with lightweight computation, avoiding dense matrix multiplication over all token pairs.

**Anchor Sampling and Sub-matrix Alignment.** We first perform uniform anchor sampling on the Query and Key sequences to obtain a subset of indices $I_{sub}$ (where $|I_{sub}| = m \ll N$). We use uniform sampling by default because Stage 2 is designed as a lightweight coefficient-estimation step; more complex adaptive sampling requires per-layer scoring and sorting, which can erode the practical speedup. Using these anchor tokens, we compute a miniature "Anchor Interaction Matrix" $S_{sub} \in \mathbb{R}^{m \times m}$:

$$S_{sub} = Q[I_{sub}] \cdot K[I_{sub}]^T. \tag{1}$$

Simultaneously, to ensure strict spatial correspondence, we construct the basis sub-region $B_{k,sub}$ by slicing the pre-stored basis $B_k$ using the identical row and column indices $I_{sub}$, i.e., $B_{k,sub} = B_k[I_{sub}, :][:, I_{sub}]$.

**Orthogonal Projection Solver.** Leveraging the orthogonality of the learned bases, we formulate the weight estimation as a local projection problem. The coefficient $w_k$ for the $k$-th basis is estimated by projecting the sampled interaction $S_{sub}$ onto the corresponding basis sub-region:

$$w_k \approx \frac{\langle S_{sub}, B_{k,sub} \rangle}{\|B_{k,sub}\|_F^2}, \tag{2}$$

where $\langle \cdot, \cdot \rangle$ denotes the Frobenius inner product, and $\| \cdot \|_F$ is the Frobenius norm. This formulation assumes that the energy distribution within the sampled sub-region effectively approximates the dominant structure of the full attention interaction.

**Linear Reconstruction and Complexity Analysis.** Upon obtaining the weights $\mathbf{w} = \{w_1, ..., w_K\}$, we reconstruct the approximate attention matrix via linear combination:

$$S_{approx} = \sum_{k=1}^{K} w_k B_k. \tag{3}$$

**Computational Advantage:** Standard self-attention requires $d$ floating-point multiplications, typically $d = 128$, for every element in the $N \times N$ map to compute dot products. In contrast, Video-SVD transforms this expensive vector operation into a scalar operation: we only need to perform a

weighted sum of $K$ bases, typically $K = 16 \ll d$. Furthermore, since the projection in Eq. 2 operates only on $m \times m$ sampled anchors, the estimation cost remains lightweight, reducing the overall attention bottleneck significantly.

### 3.3. Stage 3: Structured Error-Aware Compensation Mechanism

Low-dimensional reconstruction captures the dominant semantic structure of attention but cannot fully recover all fine-grained content details and positional information. To address this gap, we design a Structured Error-Aware Compensation Mechanism that explicitly predicts residual corrections on top of the reconstructed attention output.

**Global Error Descriptor.** To guide the compensation network, we first quantify the reconstruction error within the anchor region:

$$D_{sub} = S_{sub} - \sum_{k=1}^{K} w_k B_{k,sub}. \tag{4}$$

We then compress this error into a conditional prompt $\mathbf{v}_{err} \in \mathbb{R}^{d_{err}}$ via Adaptive Average Pooling:

$$\mathbf{v}_{err} = \text{Flatten}(\text{AvgPool}(D_{sub})). \tag{5}$$

**Dual-Stream Compensation Architecture.** We adopt a parallel dual-stream architecture for feature recalibration:

- **Content Stream** ($R(\mathbf{content})_i$)**:** Targeting fine-grained content and texture details, this MLP uses the Value matrix $V$, broadcasted weights $\mathbf{w}$, and the error descriptor $\mathbf{v}_{err}$ to predict content-dependent residuals:

$$R(\text{content})_i = \text{MLP}_{content}([V \oplus \mathbf{w} \oplus \mathbf{v}_{err}]). \tag{6}$$

- **Position Stream** ($R(\mathbf{Pos})_i$)**:** Targeting positional information, this shared MLP leverages sinusoidal RoPE frequencies as a **structural prior** to recover positional residuals:

$$R(\text{Pos})_i = \text{MLP}_{pos}(\text{RoPE}_{freq}). \tag{7}$$

**Final Output and Optimization.** The final output aggregates the Video-SVD approximation and compensation terms:

$$\begin{aligned} \text{Head}_i \text{ Output} = \text{softmax}(S_{approx}^{(i)}) \cdot V^{(i)} \\ + R(\text{content})_i + R(\text{Pos})_i. \end{aligned} \tag{8}$$

During training, we freeze the offline bases and minimize the Frobenius norm distance between the final output and the ground-truth attention output $O_{GT}$:

$$\mathcal{L}_{rec} = \sum_i \|O_{GT}^{(i)} - \text{Head}_i \text{ Output}\|_F^2. \tag{9}$$

*Table 1.* **Quantitative comparison on state-of-the-art VDiTs.** Evaluated on HunyuanVideo and Wan2.1 benchmarks, our method (highlighted in gray) achieves a superior efficiency-quality trade-off while significantly reducing memory consumption.

| Model | Method | Quality Metrics | | | | Efficiency Metrics | | |
|---|---|---|---|---|---|---|---|---|
| | | VBench ↑ | PSNR ↑ | SSIM ↑ | LPIPS ↓ | Latency (s) ↓ | Speedup ↑ | Mem. (GB) ↓ |
| **HunyuanVideo** (128 frames) | *Original* | 82.35 | 29.9 | 0.965 | 0.105 | 1095 | 1.00× | 47.6 |
| | STA | 81.95 | 22.7 | 0.821 | 0.327 | 730 | 1.50× | 51.8 |
| | PAB | 82.10 | 23.3 | 0.752 | 0.175 | 715 | 1.53× | >80 |
| | VORTA | 82.25 | 25.9 | 0.837 | 0.185 | 622 | 1.76× | 51.2 |
| | SVG | 82.28 | 26.4 | 0.834 | 0.163 | 633 | 1.73× | 48.5 |
| | **Ours** | **82.32** | **29.5** | **0.917** | **0.126** | **570** | **1.92×** | **40.5** |
| **Wan2.1-1.3B** (81 frames) | *Original* | 81.80 | 27.9 | 0.950 | 0.115 | 469 | 1.00× | 25.5 |
| | STA | 81.05 | 22.1 | 0.805 | 0.280 | 323 | 1.45× | 26.1 |
| | PAB | 81.35 | 23.7 | 0.790 | 0.181 | 316 | 1.48× | 35.0 |
| | VORTA | 81.65 | 24.8 | 0.815 | 0.176 | 302 | 1.55× | 27.2 |
| | SVG | 81.72 | 25.2 | 0.812 | 0.179 | 296 | 1.58× | 26.0 |
| | **Ours** | **81.76** | **27.5** | **0.825** | **0.151** | **268** | **1.75×** | **21.8** |
| **Wan2.1-14B** (81 frames) | *Original* | 82.49 | 28.2 | 0.960 | 0.108 | 1153 | 1.00× | 42.8 |
| | STA | 81.90 | 24.8 | 0.882 | 0.208 | 854 | 1.35× | 43.5 |
| | PAB | 82.10 | 25.6 | 0.880 | 0.210 | 823 | 1.40× | >60 |
| | VORTA | 82.35 | 26.0 | 0.868 | 0.224 | 758 | 1.52× | 45.0 |
| | SVG | 82.38 | 27.1 | 0.888 | 0.212 | 729 | 1.58× | 43.2 |
| | **Ours** | **82.43** | **28.0** | **0.892** | **0.188** | **644** | **1.79×** | **37.3** |

## 4. Experiments

**Models** We benchmark our method on two popular diffusion models: HunyuanVideo (Kong et al., 2024), and Wan2.1 (Wan et al., 2025), which have 13, 1.3, and 14 billion parameters, respectively. HunyuanVideo can generate up to video with 720p resolution and 128 frames. Wan2.1-14B can generate up to video with 720p and 81 frames.

**Datasets** For text-to-video generation, we use exactly 600 prompts from the Penguin Benchmark after prompt optimization by VBench (Huang et al., 2024). For each evaluated model, we generate exactly 600 videos under the same prompt set. These prompts cover diverse semantic categories, including subject dynamics such as complex animal behaviors and detailed human actions, spatial relationships involving relative positioning of multiple objects, camera movements such as medium-long shots, wide-angle shots, and explicit panning, and visual styles such as 4K realistic videography, 3D cartoon, sci-fi art, and anime. This 600-prompt evaluation set is separate from the 600-video calibration pool used for basis extraction.

**Metrics** Our evaluation covers both quality and efficiency. For quality, we compare against a full attention model using PSNR, LPIPS, and SSIM, while VBench (Huang et al., 2024) assesses background consistency and motion smoothness. For efficiency, we measure the sparse attention cost with density and the generation cost with total FLOPs.

**Baselines** We compare against state-of-the-art sparse attention algorithms including static method PAB(Zhao et al., 2025), STA(Zhang et al., 2025b), VORTA(Zhang et al., 2025b) and Sparse VideoGen (SVG)(Xi et al., 2025).

### 4.1. Quality Evaluation

To assess the generation quality, we evaluate the quality of videos generated by Video-SVD against baselines, as shown in Table 1 and Fig. 5. Video-SVD consistently outperforms all baseline methods (STA, PAB, VORTA, SVG) in terms of PSNR, SSIM, and LPIPS. Specifically, Video-SVD achieves high PSNR scores of 29.5 on HunyuanVideo, 27.5 on Wan2.1-1.3B, and 28.0 on Wan2.1-14B. Beyond quality, our method demonstrates superior efficiency, providing significant speedups of 1.92×, 1.75×, and 1.79× respectively. Crucially, Video-SVD also optimizes memory efficiency; for instance, it reduces the memory footprint of Wan2.1-1.3B to 21.8 GB(For a more detailed mathematical derivation and implementation specifics, please refer to **Appendix A**.), enabling deployment on consumer-grade GPUs. Furthermore, as shown in Fig. 5, Video-SVD maintains high fidelity across diverse motion complexities, gracefully handling high-motion scenarios (e.g., a potter's wheel, a chef kneading dough). Notably, in the chef kneading sequence, our method preserves distinct finger details and a sharp boundary between the hands and the dough. In contrast, baselines like SVG struggle with such rapid motion, causing the hands and dough to blur together indistinguishably.

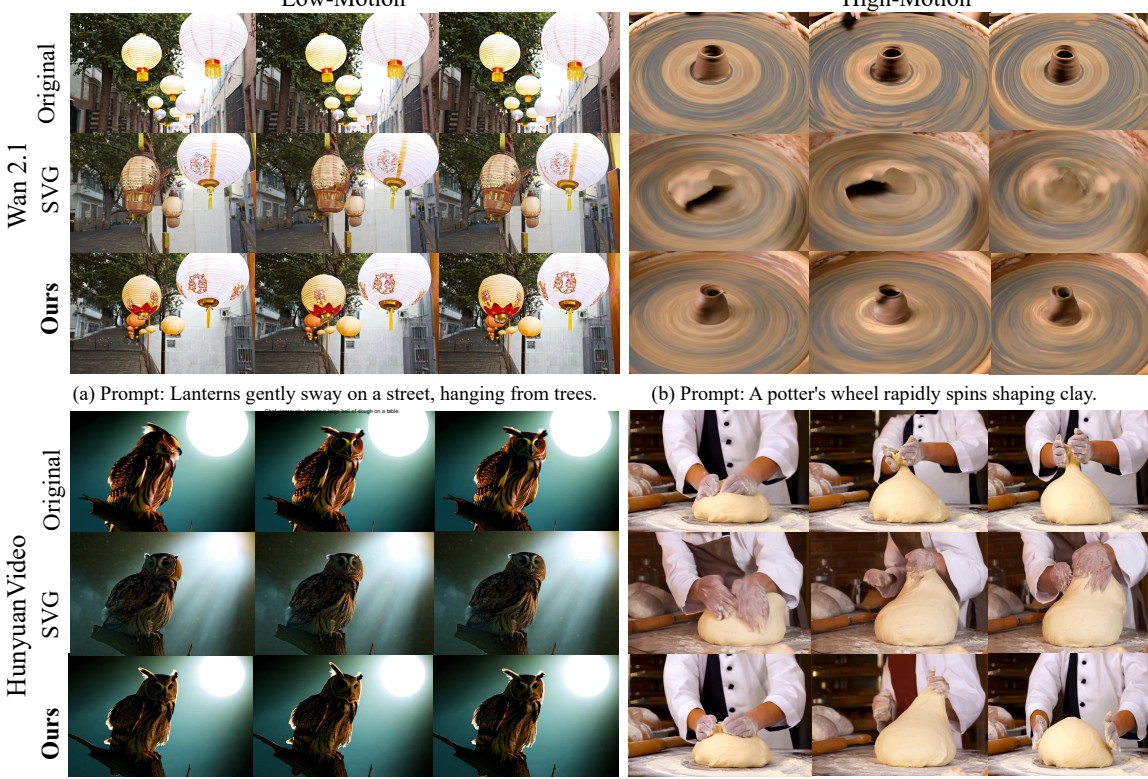

(a) Prompt: Lanterns gently sway on a street, hanging from trees.

(b) Prompt: A potter's wheel rapidly spins shaping clay.

(c) Prompt: Owl perches on a branch turning head under moonlight.

(d) Prompt: Chef vigorously kneads a ball of dough on a table.

*Figure 5.* **Qualitative Comparison.** Video-SVD achieves high fidelity across varying motion complexities. While SVG suffices for low-motion videos (a, c), it suffers from severe distortion in high-motion scenarios like the spinning wheel and kneading chef (b, d). This highlights the limitation of SVG's rigid patterns compared to our robust, data-driven global basis which effectively handles complex dynamics.

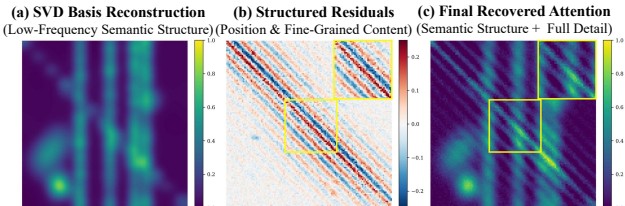

*Figure 6.* **Visualization of frequency-aware attention reconstruction.** (a) SVD basis captures global low-frequency semantic structures. (b) Structured residuals recover high-frequency positional information (e.g., diagonal RoPE textures) and fine-grained content. (c) Final recovered attention achieves high-fidelity detail restoration.

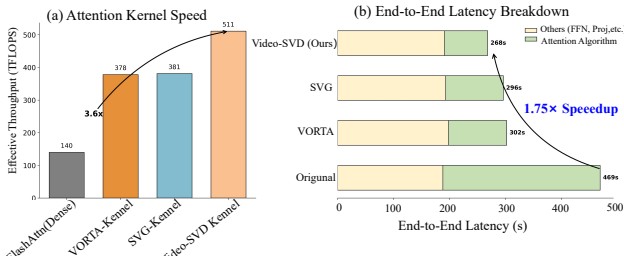

*Figure 7.* **Efficiency and Latency.** (a) Our Video-SVD kernel achieves 511 TFLOPS, a 3.6× speedup over FlashAttn. (b) This yields the fastest latency of 268s (1.75× speedup), where gains derive solely from the optimized Attention Algorithm (green) rather than other constant costs (yellow).

### 4.2. Ablation Study

We ablate Video-SVD's core components in Table 2, where the "Basis only" configuration yields the peak 1.95× speedup but suffers a significant quality drop to 68.50 VBench. Incorporating either $R_{RoPE}$ or $R_{content}$ partially restores performance, confirming that both positional and content residuals are essential for high-fidelity reconstruction. Finally, our full model (Full) effectively matches the original quality (81.76 vs. 81.80 VBench) while maintain-

ing a substantial 1.75× speedup and a reduced 21.8 GB memory footprint.

Fig. 6 illustrates the frequency-aware decomposition of Video-SVD. The SVD basis (Fig. 6a) reconstructs the low-frequency semantic skeleton, but its low-pass nature fails to model high-frequency positional oscillations. Our dual-stream structured residuals (Fig. 6b) effectively recover these RoPE-induced textures and content details, restoring full-attention fidelity (Fig. 6c).

*Table 2.* **Ablation study on the core components of Video-SVD.** We evaluate the contribution of the low-rank basis and residual terms ($R_{RoPE}$ and $R_{content}$) on the Wan2.1-1.3B model. The table reports the impact of each component on both generation quality (VBench, PSNR, etc.) and inference efficiency (FLOPs, Speedup).

| Method | Quality Metrics | | | | Efficiency Metrics | | | |
|---|---|---|---|---|---|---|---|---|
| | VBench ↑ | PSNR ↑ | SSIM ↑ | LPIPS ↓ | FLOPs (T) ↓ | Latency (s) ↓ | Speedup ↑ | Mem. (GB) ↓ |
| Original (Full Attention) | 81.80 | 27.9 | 0.950 | 0.115 | 560 | 469 | 1.00× | 25.5 |
| Video-SVD (Basis only) | 71.53 | 21.1 | 0.680 | 0.350 | **235** | **240** | **1.95×** | **20.5** |
| Video-SVD (Basis + $R_{RoPE}$) | 76.51 | 23.4 | 0.790 | 0.280 | 238 | 246 | 1.91× | 20.8 |
| Video-SVD (Basis + $R_{content}$) | 78.24 | 25.8 | 0.760 | 0.240 | 255 | 258 | 1.82× | 21.3 |
| **Video-SVD (Ours, Full)** | **81.76** | **27.5** | **0.825** | **0.151** | 262 | 268 | 1.75× | 21.8 |

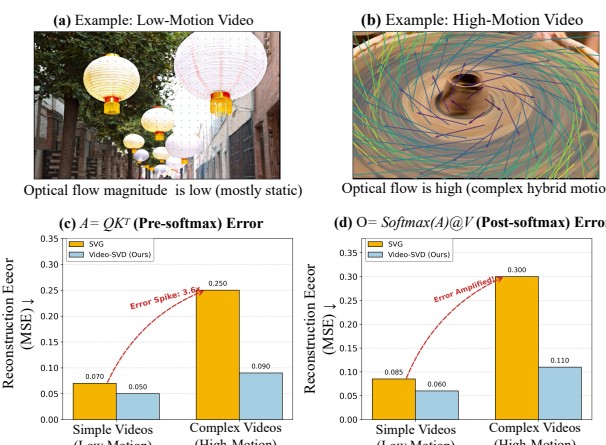

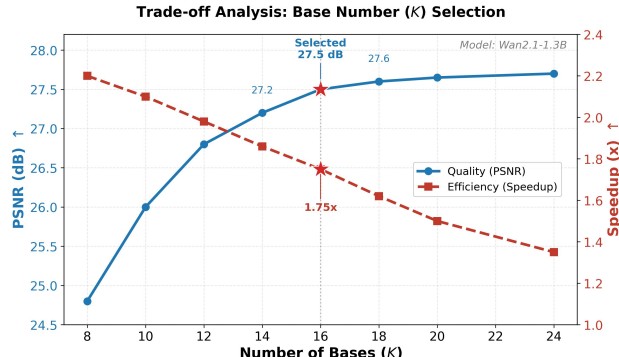

*Figure 9.* **Trade-off Analysis: Base Number ($K$) Selection.** Increasing $K$ improves quality (blue) but reduces efficiency (red). We identify $K = 16$ as the optimal operating point, achieving 27.5 dB PSNR while maintaining a 1.75× speedup.

*Figure 8.* **Reconstruction Error vs. Video Complexity.** (a) and (b) visually define our "Low-Motion" (mostly static) and "High-Motion" (complex hybrid) categories. (c) At the Pre-Softmax ($A = QK^T$) level, SVG suffers a dramatic 3.6× error spike on complex videos, whereas Video-SVD maintains robustly low error (0.090). (d) This discrepancy is further amplified at the Post-Softmax ($O = \text{softmax}(A)@V$) output, demonstrating Video-SVD's superior fidelity in handling complex motion.

### 4.3. Analysis of Efficiency and Robustness

We first analyze the source of Video-SVD's speedup. As illustrated in Fig. 7, the performance gain stems principally from our optimized Attention Algorithm. Unlike methods relying on custom CUDA kernels, Video-SVD employs checkpoint-adaptive bases to approximate the QK matrix, combined with online projection weights and an MLP to compensate for residual errors. This design allows for highly efficient computation that avoids full matrix materialization. Fig. 7(a) demonstrates that this approach achieves an effective throughput of 511 TFLOPS, significantly outperforming the equivalent throughput of VORTA and SVG. This algorithmic efficiency translates to end-to-end gains; as shown in Fig. 7(b), Video-SVD reduces total latency to 268s, achieving a 1.75× speedup compared to the Original model (469s).

Next, we investigate the fidelity of these approximations using Reconstruction Error (Fig. 8), which we posit is a more fundamental metric than final visual scores as it isolates

the attention mechanism's accuracy before subsequent layer compensation. We categorize inputs into Low-Motion and High-Motion (Fig. 8(a,b)) to test robustness. The results in Fig. 8(c) reveal a critical limitation in SVG: on complex high-motion videos, SVG's pre-softmax error spikes by 3.6× to 0.250. In contrast, Video-SVD maintains a low error of 0.090. This trend persists in the post-softmax output (Fig. 8(d)), where SVG's error amplifies to 0.300, while Video-SVD remains stable at 0.110. This analysis confirms that Video-SVD successfully delivers SOTA speed (Fig. 7) while maintaining high-fidelity attention reconstruction under complex motion (Fig. 8). Sensitivity analysis regarding sampling ratio and the synergistic robustness provided by Stage 3 compensation can be found in **Appendix B**.

### 4.4. Low-Rank and Linear Attention Baselines

To clarify the distinction between Video-SVD and generic low-rank or linear attention approximations, we compare representative drop-in baselines on Wan2.1-1.3B under the same 81-frame attention-replacement setting. As shown in Table 3, these methods can obtain comparable speedups, but they suffer substantially larger fidelity degradation. This confirms that the contribution of Video-SVD is not simply using SVD, but making low-dimensional attention reconstruction practical through checkpoint-adaptive bases,

*Table 3.* **Comparison with low-rank and linear attention baselines** on Wan2.1-1.3B. Generic approximations achieve speedup but degrade fidelity.

| Method | Latency ↓ | Speedup ↑ | PSNR ↑ | SSIM ↑ | LPIPS ↓ |
|---|---|---|---|---|---|
| Original Full Attention | 469 | 1.00× | 27.9 | 0.950 | 0.115 |
| Linformer | 257 | 1.82× | 20.8 | 0.675 | 0.358 |
| SANA-inspired Linear Attn | 246 | 1.91× | 17.4 | 0.542 | 0.406 |
| Nyströmformer | 298 | 1.57× | 23.6 | 0.748 | 0.282 |
| Video-SVD Basis Only | 240 | 1.95× | 21.1 | 0.680 | 0.350 |
| **Video-SVD Full System** | **268** | **1.75×** | **27.5** | **0.825** | **0.151** |

*Table 4.* **Sampling strategy ablation** on Wan2.1-1.3B. Uniform sampling offers the best efficiency-quality trade-off.

| Sampling | Stage-2 ↓ | Latency ↓ | Pre-Err. ↓ | Post-Err. ↓ | PSNR ↑ | LPIPS ↓ |
|---|---|---|---|---|---|---|
| Random 5% | 0.8 | 268.3 | 0.072 | 0.088 | 27.46 | 0.152 |
| Uniform 5% | 0.5 | 268.0 | 0.070 | 0.085 | 27.50 | 0.151 |
| Adaptive Top-5% | 16.1 | 283.6 | 0.068 | 0.083 | 27.53 | 0.149 |

online projection, and structured residual compensation.

## 4.5. Sampling Strategy Analysis

Stage 2 performs lightweight coefficient estimation, so the sampling strategy itself must remain cheaper than the dense attention computation it replaces. We compare random sampling, uniform sampling, and a content-aware adaptive sampler on Wan2.1-1.3B, fixing $K = 16$, the sampling ratio to 5%, and all other components unchanged. As shown in Table 4, adaptive sampling slightly improves quality but introduces substantial overhead from per-layer scoring and Top-K sorting. Uniform sampling therefore provides the best efficiency-quality trade-off at our default setting.

## 4.6. Offline Calibration Cost and Reusability

Video-SVD requires a one-time checkpoint-adaptive calibration step to construct the basis bank. This cost is amortized over all subsequent inference under the same checkpoint and resolution bucket. As shown in Table 5, streaming Incremental SVD compresses large dense attention collections into compact reusable basis banks, making the offline cost practical compared with storing dense matrices directly.

## 4.7. Impact of Basis Number on Quality and Efficiency

To determine the default basis number $K$, we conduct a trade-off analysis on Wan2.1-1.3B, as shown in Fig. 9. Increasing $K$ improves reconstruction quality but reduces inference speedup. When $K$ is small (e.g., $K = 8$), Video-SVD achieves higher speedup but suffers from insufficient fidelity. Increasing $K$ to 16 significantly improves PSNR to 27.5 dB while maintaining a 1.75× speedup. Further increasing $K$ to 24 yields only marginal quality gains but introduces a substantial speed penalty. We therefore set $K = 16$ as the default configuration, which provides the best quality-efficiency trade-off.

*Table 5.* **Offline calibration budget.** One-time basis extraction cost using streaming Incremental SVD.

| Model | #Mat. | Protocol | Collect | SVD | Storage |
|---|---|---|---|---|---|
| Wan2.1-1.3B | 10,800 | 30L×60V×3T×2H | 18h GPU | 75h CPU | 23GB vs. 17.2TB |
| Wan2.1-14B | 19,200 | 40L×80V×3T×2H | 38h GPU | 130h CPU | 45GB vs. 30.7TB |

*Table 6.* **Zero-shot transfer under dataset shift.** Bases and residual modules are calibrated only on VBench and directly applied to unseen datasets.

| Dataset | Full-Attn PSNR | Video-SVD PSNR | Drop |
|---|---|---|---|
| UCF-101 | 28.52 | 28.15 | -0.37 |
| DEVIL | 27.24 | 26.71 | -0.53 |
| EvalCrafter | 26.85 | 26.38 | -0.47 |
| Average Drop | – | – | -0.46 |

## 4.8. Transferability and Boundary Analysis

We further evaluate Video-SVD under dataset shift and analyze its boundary cases by calibrating the offline bases and residual modules only on VBench and directly applying them to unseen datasets. As shown in Table 6, Video-SVD exhibits only modest degradation across UCF-101, DEVIL, and EvalCrafter, with an average PSNR drop of 0.46 dB. We observe no structural collapse or severe artifacts, suggesting that the learned bases capture reusable low-dimensional structural motifs across diverse video domains.

Video-SVD generalizes well when the dominant attention structure remains compressible and aligned with the learned basis bank. However, highly non-natural, high-frequency synthetic domains such as *Voxel* or *Pixel Art* remain challenging, where low-dimensional reconstruction can smooth sharp discontinuities. We use the projection residual as a runtime diagnostic and selectively route difficult heads back to exact FlashAttention. In our high-frequency OOD experiment, routing only 3% of the hardest heads improves PSNR from 17.5 dB to 23.9 dB while preserving a 1.71× end-to-end speedup. See **Appendix C** for detailed analysis.

## 5. Conclusion

We present Video-SVD, a plug-and-play framework for accelerating Video Diffusion Transformers with checkpoint-adaptive basis composition and structured residual compensation. By exploiting the effective low-dimensional structure and hybrid spatio-temporal motifs of video attention, Video-SVD avoids rigid sparse pruning while preserving dense global context. Experiments on HunyuanVideo and Wan2.1 show strong speedups, memory reduction, and high generation fidelity, especially under complex motion. Further analyses on linear baselines, sampling, calibration, and transferability confirm the robustness, practicality, and broad applicability of our design.

## Acknowledgement

This work was supported by the Institute Innovation Project of the Institute of Computing Technology, Chinese Academy of Sciences under Grant No. E561090.

## Impact Statement

This work improves the efficiency of video diffusion models, reducing inference cost and memory usage while maintaining high generation fidelity. More efficient video generation may broaden access to generative models and reduce resource consumption. At the same time, we acknowledge misuse risks of synthetic video generation, and encourage responsible deployment, disclosure, and safeguards such as detection and watermarking.

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

# Appendix

## A. Memory Optimization: From Basis Storage to Tiled Inference

In this section, we provide a step-by-step technical analysis of the memory optimization pipeline in Video-SVD, detailing how we achieve a linear scaling of VRAM usage relative to sequence length.

### A.1. Phase 1: SVD-based Basis Extraction

To eliminate the massive redundancy in the $QK^T$ matrix, we perform the following offline steps:

- **Full Factorization**: For any dense $QK^T$ matrix $A \in \mathbb{R}^{N \times N}$, SVD yields $A = U\Sigma V^T$, capturing the full manifold structure.

- **Energy Concentration**: We observe that the singular values in $\Sigma$ are highly sparse, with dominant values capturing essential spatiotemporal features like motion trajectories and global context.

- **Optimal Truncation**: Following the **Eckart-Young theorem**, we retain only the top $K$ singular values and their corresponding vectors to construct low-rank bases $\tilde{U}, \tilde{V} \in \mathbb{R}^{N \times K}$. This truncation provides the optimal rank-$K$ approximation of the original attention prior with minimal fidelity loss.

### A.2. Phase 2: Storage Complexity and Basis Sharing

The transition to low-rank bases fundamentally alters the memory footprint:

- **Quadratic to Linear Transition**: For the $QK^T$ matrix, the storage complexity shifts from $O(N^2)$ to $O(NK)$ . This shift ensures that the memory requirement for attention priors remains manageable as the number of tokens $N$ increases.

- **Shared Basis Mechanism**: We implement a cross-layer sharing scheme that further reduces the static memory overhead of learned bases, preventing a linear explosion of parameters as the model depth increases.

- **Efficiency Gain**: In high-resolution video synthesis where $N$ often exceeds $10^4$, this optimization leads to a storage reduction of multiple orders of magnitude compared to dense maps.

### A.3. Phase 3: Tiled Inference and Peak Memory Suppression

To prevent $O(N^2)$ memory peaks during the forward pass, we implement a "calculate-and-evict" strategy:

1. **Tile-wise Reconstruction**: We partition the query space into tiles of size $N_{tile}$. In each iteration, we only compute the local interaction between $\tilde{U}_{tile}$ and the global basis $\tilde{V}^T$.

2. **On-the-fly Materialization**: The resulting $N_{tile} \times N$ partial weights are passed through the softmax and immediately consumed by the value-matrix multiplication.

3. **Immediate VRAM Eviction**: Crucially, the intermediate $N_{tile} \times N$ tensor is **immediately released** from the GPU memory after each step.

This ensures that peak VRAM consumption is bounded by $O(N \cdot N_{tile})$ rather than the sequence length squared, facilitating the deployment of large-scale Video Diffusion Transformers on consumer hardware.

## B. Detailed Sensitivity and Robustness Analysis

In this section, we provide a comprehensive analysis of the anchor sampling strategy employed in Stage 2 of Video-SVD. We verify both the stability of the subspace projection and its resilience across varying temporal dynamics.

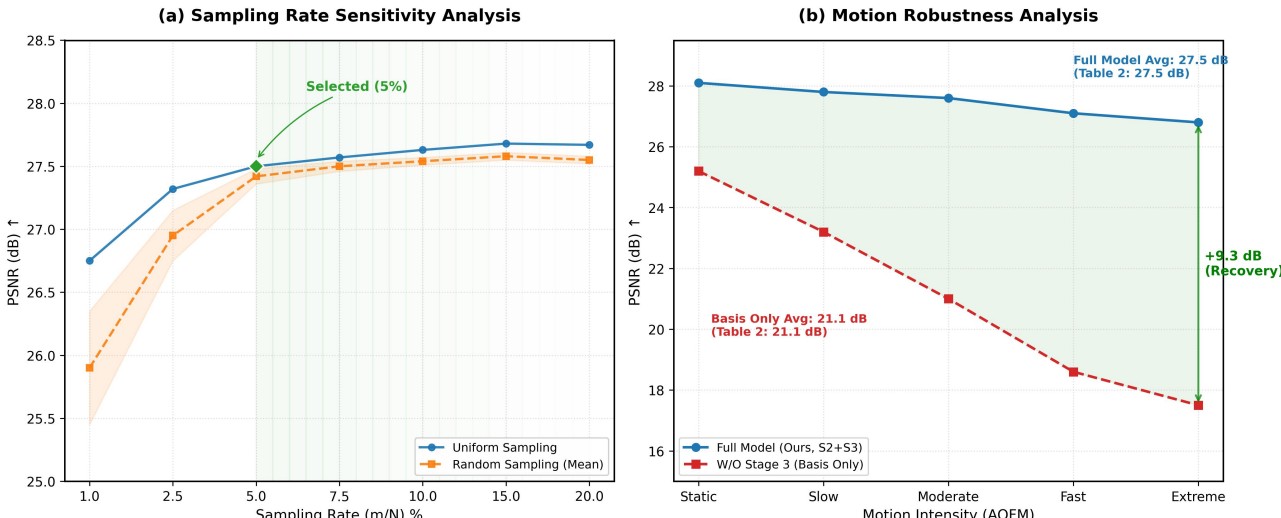

*Figure 10.* Anchor Sampling Analysis. (a) Reconstruction quality stabilizes at a 5% sampling ratio with negligible variance. (b) Stage 3 compensation provides a robust safety net with a +9.3 dB recovery in extreme motion scenarios, ensuring high fidelity across diverse dynamics.

### B.1. Experimental Setup

To quantify the reliability of sparse sampling, we conduct sensitivity tests on the Wan2.1-1.3B model. We vary the sampling ratio $m/N$ from $1\%$ to $20\%$ to evaluate convergence properties. Furthermore, we categorize inputs based on Average Optical Flow Magnitude (AOFM) into five motion tiers—ranging from *Static* to *Extreme*—to assess reconstruction performance under pressure.

### B.2. Discussion of Results

As illustrated in Fig. 10, our method demonstrates exceptional stability:

**Stability of Anchor Sampling (Fig. 10)):** The results indicate that reconstruction quality exhibits a rapid saturation curve. Performance converges at a 5% sampling ratio, achieving a PSNR of 27.5 dB (matching the results in Table 2). The standard deviation for random sampling remains minimal ($\sigma < 0.05$ dB) in the stable region, confirming that the learned Global Bases effectively capture the dominant attention structures and are not sensitive to the specific choice of anchor tokens.

**Robustness to Motion Dynamics (Fig. 10):** While the *Basis Only* approximation degrades to 17.5 dB under extreme motion tiers, our Stage 3 Compensation provides a critical recovery of +9.3 dB. This synergy ensures that the full model maintains a high-fidelity output (averaging 27.5 dB) regardless of temporal complexity, demonstrating the robustness of our dual-stage architecture.

## C. Visualization of the Generated Videos

We provide visualization comparisons between Video-SVD and full attention on the Wan 2.1 benchmark. Results in Fig 11 and Fig 12 demonstrate that Video-SVD preserves high pixel-level fidelity, achieving generation quality comparable to that of full attention. Real video samples are provided in the supplementary materials.

In these non-natural, high-frequency domains, Video-SVD operates as a low-rank filter, inherently smoothing sharp edges and leading to degraded quality, as illustrated in Fig 13.

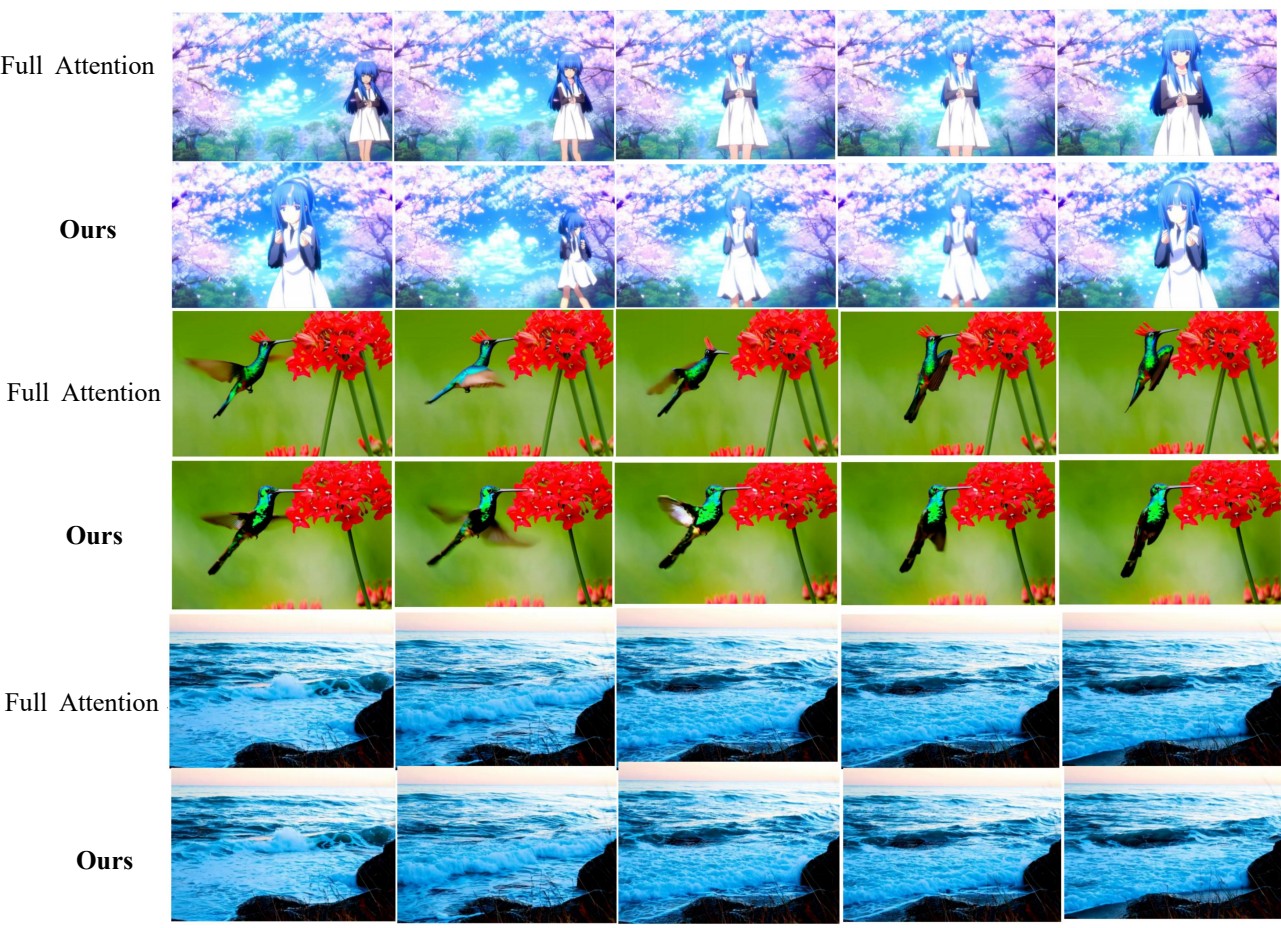

*Figure 11.* Comparion of Full Attention and Video-SVD on Wan 2.1 Text-to-Video generation.

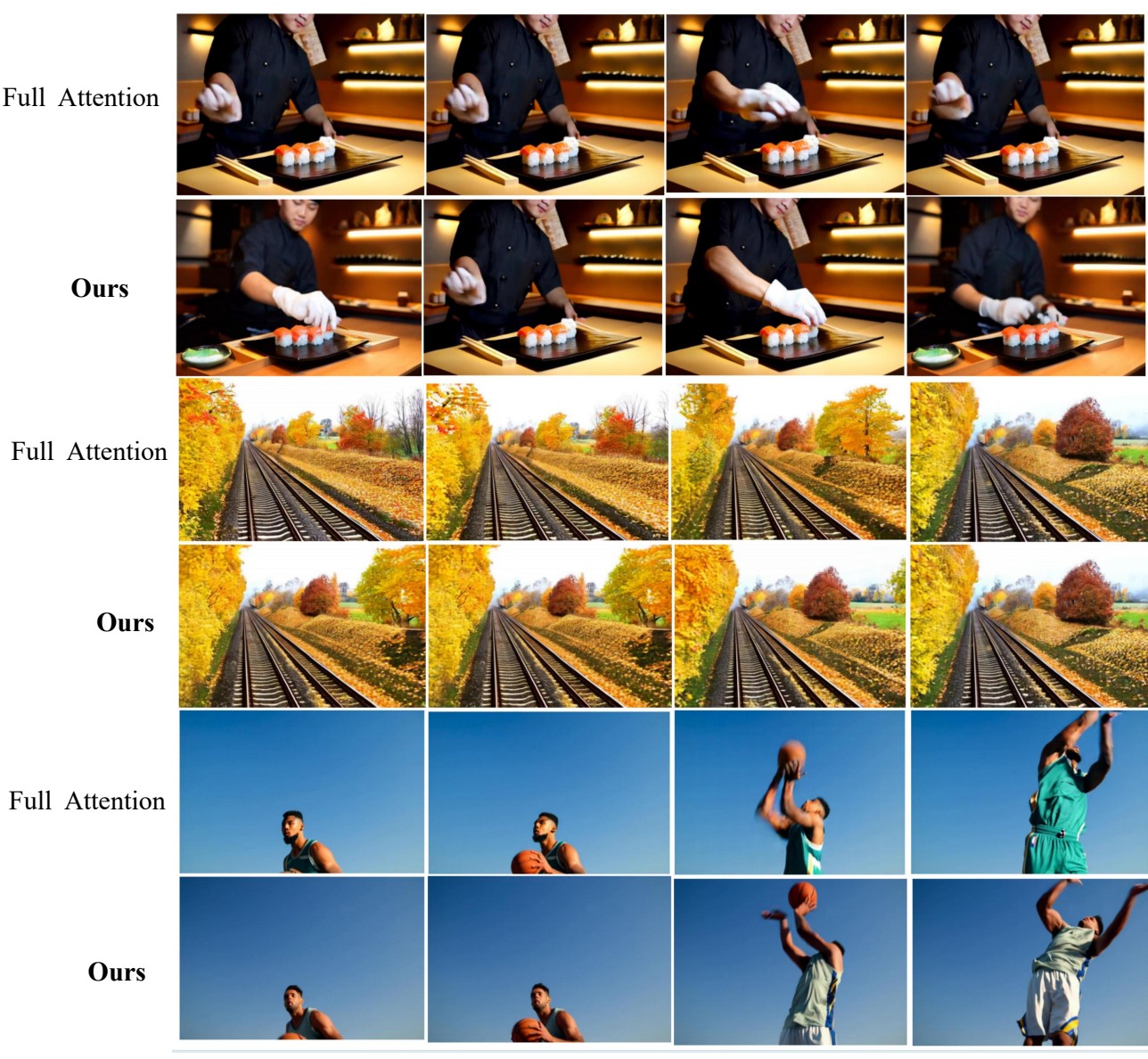

*Figure 12.* Comparion of Full Attention and Video-SVD on Wan 2.1 Text-to-Video generation.

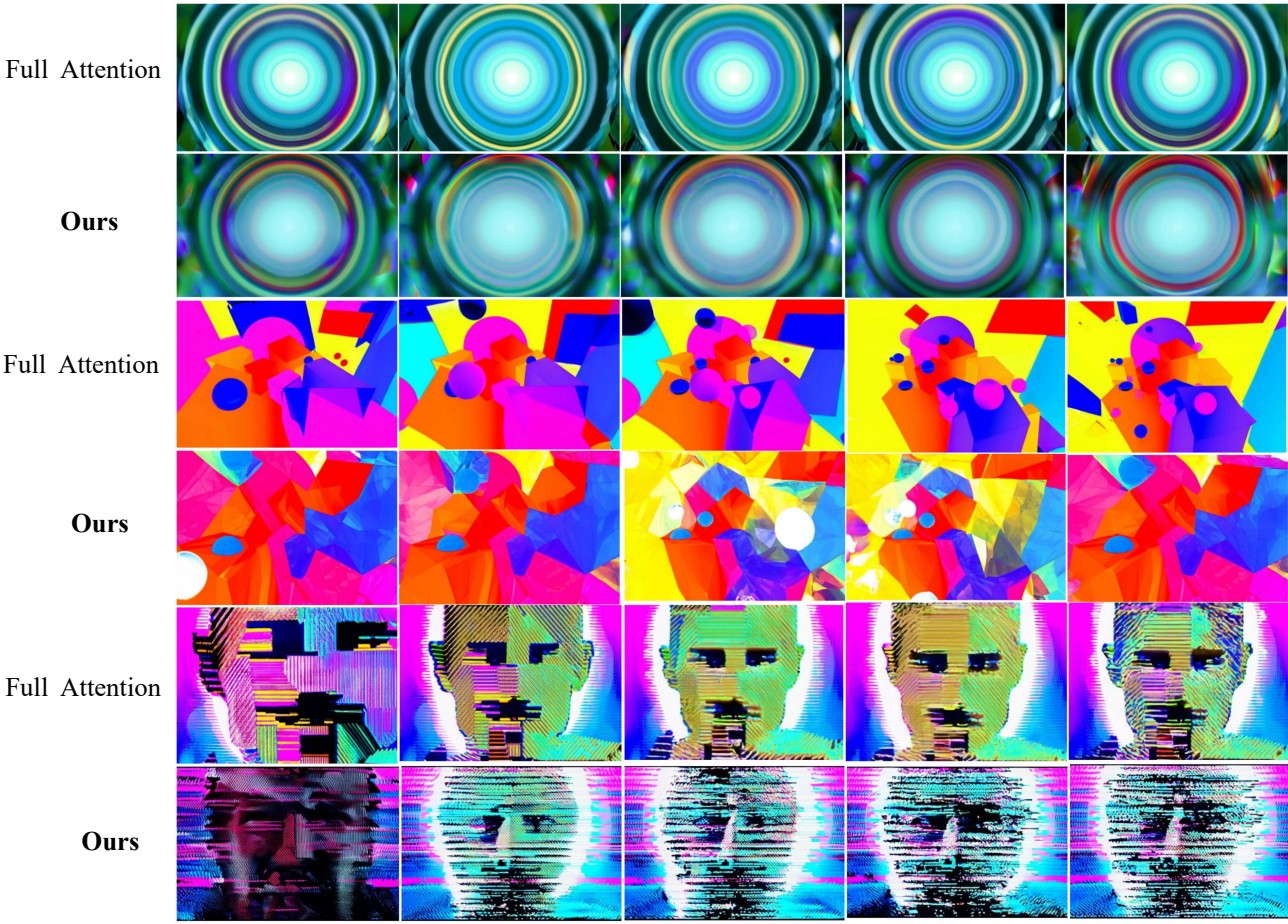

*Figure 13.* Algorithmic Boundary and OOD Analysis. In non-natural, high-frequency domains within the Wan 2.1 benchmark (e.g., pixel or voxel art), Video-SVD acts as a low-rank filter that inherently smooths sharp edges, leading to a degradation in generation quality compared to full attention.

