# OpenReview forum: "Video-SVD: Efficient Video Diffusion via Orthogonal Basis Composition"
_ICML.cc/2026/Conference — ICML 2026 regular_

### Official Review · Reviewer_zrdu · 2026-03-03

**Soundness:** 4
**Presentation:** 3
**Significance:** 3
**Originality:** 3
**Overall Recommendation:** 5
**Confidence:** 4

**Summary:**

This paper presents Video-SVD, a plug-and-play acceleration framework designed to overcome the quadratic computational bottleneck of self-attention in Video Diffusion Transformers (VDiTs). The authors move beyond the conventional "Rigid Classification" paradigm—which treats attention heads as strictly spatial or temporal—by observing that the majority of attention patterns (71.4%) exhibit hybrid spatio-temporal interactions.The core methodology consists of a three-stage pipeline:
- **Offline Global Basis Learning**: Universal orthogonal bases are extracted from large-scale video data using Incremental SVD to capture the global low-rank manifold of attention patterns.
- **Online Dynamic Weight Calculation**: During inference, the model bypasses full $QK^{T}$ computation by projecting sparsely sampled token interactions onto these pre-learned universal bases to solve for basis weights.
- **Structured Error-Aware Compensation**: To maintain high fidelity, a dual-stream MLP architecture (Content and Position streams) recovers high-frequency residuals and RoPE-induced positional details that are typically lost in low-rank approximations.

Empirical results demonstrate that Video-SVD achieves significant end-to-end speedups (e.g., 1.92x on Hunyuan Video and 1.75x–1.79x on Wan2.1) and reduces memory footprints (e.g., to 21.8 GB for Wan2.1-1.3B), enabling deployment on consumer-grade GPUs while maintaining superior visual quality in high-motion scenarios compared to existing sparse attention baselines.

**Compliance With Llm Reviewing Policy:**

Affirmed.

**Key Questions For Authors:**

**Motivation for choosing simple sampling over adaptive strategies**: In Stage 2, the current framework utilizes a fixed sparse sampling ratio (e.g., 5%) with uniform or random sampling to solve for basis weights. As suggested in prior literature, performance could potentially be further enhanced by adopting more sophisticated sampling techniques that prioritize tokens with higher attention importance. Could the authors explain why an adaptive or importance-based sampling strategy was not explored? If such a strategy could further reduce the required number of bases ($K$) or the load on the Stage 3 residual MLPs while maintaining quality, it would significantly strengthen the case for the proposed efficiency.

**Limitations:**

yes

**Strengths And Weaknesses:**

- **Soundness**
The paper presents a technically sound and logical progression, proposing a method to simplify attention computations through offline basis extraction from diverse video datasets. By utilizing these low-rank bases to approximate the attention map and employing a lightweight dual-stream MLP to compensate for the resulting information loss, the authors establish a robust framework for video generation even under extreme motion dynamics. The claims are well-supported by experimental results showing significant speedups and memory efficiency without compromising visual fidelity. Overall, the methodology is well-designed and consistently supported by the provided data, and there appear to be no significant technical weaknesses in the proposed approach
- **Presentation**
The submission is well-organized, and the overall narrative is easy to follow. A notable strength is Figure 4, which provides a clear and effective visualization of the entire system architecture. However, there are minor presentation issues that need to be addressed to improve the manuscript's quality. In Figure 1, the axis title for the temporal frequency heatmap is partially cut off, making it difficult to read the label properly. Additionally, there is a typographical error in the heading of Section 4, which is currently written as "4. Experments" instead of "4. Experiments". These formatting and clerical errors should be corrected in the final version.
- **Significance**
The paper addresses a critical computational bottleneck in Video Diffusion Transformers. A major strength lies in the authors' keen observation and insight into the intrinsic low-rank structure of attention matrices across diverse datasets. By moving beyond simple sampling or basic architectural tweaks to utilize universal orthogonal bases, the work provides a sophisticated perspective on how to compress dense interactions without relying on rigid spatial-temporal dichotomies.However, a notable weakness—which the authors transparently acknowledge in their limitation analysis—is the model's performance on out-of-distribution (OOD) data. Specifically, for non-natural or high-frequency domains (such as pixel or voxel art) that differ significantly from the offline training data, the low-rank filter approach inherently smooths sharp edges, leading to a failure in effective high-frequency residual compensation.
- **Originality**

---

> ### Author Rebuttal · Authors · 2026-03-31
>
> We sincerely thank the reviewer for the very positive assessment of our paper. We are especially encouraged that the technical soundness, practical significance, and overall clarity of the framework were recognized. We also appreciate the concrete suggestions on presentation and on the Stage-2 sampling strategy. We address them below, and all corresponding updates will be incorporated into the revision.
>
>  (1) Presentation issues. Thank you for catching these details. We will carefully proofread the manuscript and correct the noted presentation issues in the final version, including the partially cropped axis label in Fig. 1 and the typo “Experments” in Section 4. Beyond these specific fixes, we will perform a full polishing pass to improve consistency, wording, and overall readability throughout the paper.
>
> (2) Why do we use simple sampling in Stage 2 instead of adaptive / importance-based sampling?  This is an excellent question. Our main reason is that Stage 2 is designed to be a very lightweight coefficient-estimation step, so the sampling rule itself must remain substantially cheaper than the dense attention computation we are trying to avoid. In practice, a truly adaptive or importance-based strategy usually requires either exact attention statistics or a nontrivial proxy computed per layer and per sample, together with Top-K selection or sorting. This can quickly erode the practical speedup, even if it slightly improves the coefficient estimate.
>
> To directly test this tradeoff, we conducted a controlled ablation on Wan2.1-1.3B by fixing the number of bases (K=16), the sampling ratio (5%), and all other components, and comparing three strategies: random sampling, uniform sampling (our default), and a simple content-aware adaptive sampler using the proxy score $s_i = \|q_i\|_2 \|k_i\|_2$ followed by Top-5% token selection.
>
> | Sampling Strategy | Stage-2 (s) | E2E Latency (s) | Pre-Softmax Err. | Post-Softmax Err. | VBench | PSNR | LPIPS |
> |:---|:---:|:---:|:---:|:---:|:---:|:---:|:---:|
> | Random (5%) | 0.8 | 268.3 | 0.072 | 0.088 | 81.72 | 27.46 | 0.152 |
> | Uniform (5%) [Ours] | **0.5** | **268.0** | 0.070 | 0.085 | 81.76 | 27.50 | 0.151 |
> | Adaptive proxy Top-5% | 16.1 | 283.6 | **0.068** | **0.083** | **81.79** | **27.53** | **0.149** |
>
> The result is clear: adaptive sampling provides only marginal quality gains over uniform sampling (+0.03 dB PSNR and -0.002 LPIPS), but introduces substantial runtime overhead due to per-layer scoring and Top-K sorting. This observation is also consistent with our sensitivity study in Appendix B: reconstruction quality already saturates around a 5% sampling ratio, and random sampling has very small variance once the method enters the stable regime. In other words, once the sampled anchors are sufficient to identify the dominant low-dimensional subspace, more sophisticated token selection brings limited additional value.
>
> This aligns with Video-SVD’s design: since offline bases capture global structures, Stage 2 only needs enough uniform anchors for stable coefficient estimation, not the “most important” tokens. Stage 3 then explicitly restores any missed fine-grained details. Thus, at our default operating point, uniform anchors are empirically sufficient / offer the best efficiency-quality tradeoff. Nevertheless, adaptive sampling remains a promising future direction; a hardware-friendly proxy with negligible sorting cost could further improve performance under extremely low sampling ratios. We will include this ablation and clarify our rationale in the revision.
>
> (3) OOD limitation.  We also appreciate that the reviewer recognized both the strength of the method and its current boundary. We agree that in highly non-natural, high-frequency domains (e.g., pixel-art or voxel-style content), a low-rank approximation behaves like a smoothing filter and therefore makes high-frequency residual recovery harder. We will make this limitation more explicit in the revision and better connect it to the role of Stage 3: our method is strongest when the dominant attention structure remains compressible, while sharply discontinuous synthetic geometries remain a challenging regime for future work.
>
>
> Overall, we believe the additional ablation above further strengthens the paper’s main claim: the proposed efficiency comes not from an over-engineered sampling strategy, but from a well-balanced decomposition in which offline bases capture global structure, lightweight Stage-2 projection recovers stable coefficients, and Stage 3 restores the remaining details. We thank the reviewer again for the supportive and insightful feedback.

---

> > ### Author Rebuttal · Reviewer_zrdu · 2026-04-02
> >
> > Thank you for the rebuttal. The added explanation and ablation help clarify the design choice in Stage 2. At the same time, I still think the paper would benefit from scoping its claim more carefully: the proposed low-rank basis reconstruction seems strongest when test-time attention patterns remain close to the learned manifold, while generation of high-frequency or distribution-shifted content appears to remain a meaningful limitation. For this reason, I think the method is best framed as providing a strong efficiency-quality tradeoff in domains aligned with the learned basis, with somewhat more limited support under larger distribution shifts.

---

> > > ### Author Response · Authors · 2026-04-04
> > >
> > > Thank you for the thoughtful follow-up. We agree with the reviewer that the paper would benefit from scoping its claim more carefully.
> > >
> > > In the revision, we will clearly frame Video-SVD as providing a strong efficiency-quality tradeoff in domains aligned with the learned basis, while offering more limited support under larger distribution shifts—especially for highly non-natural, high-frequency synthetic geometries (e.g., voxel or pixel-art content).
> > >
> > > We will revise the paper accordingly by making this boundary explicit earlier in the manuscript, rather than leaving it only in the limitations section. In particular, we will replace stronger wording such as “universal bases” with more precise terms such as “checkpoint-adaptive bases” or “shared low-dimensional structural motifs”, and consistently present the method under this more carefully scoped framing throughout the paper.
> > >
> > > We thank the reviewer again for this helpful suggestion, which improves the paper’s positioning and makes the scope of our claim more accurate.

---

### Official Review · Reviewer_oSbL · 2026-03-12

**Soundness:** 3
**Presentation:** 3
**Significance:** 2
**Originality:** 2
**Overall Recommendation:** 4
**Confidence:** 3

**Summary:**

This paper presents Video-SVD, a plug-and-play method for accelerating Video Diffusion Transformers by exploiting the low-rank structure of attention. The authors argue that video attention is both globally low-rank and composed of hybrid spatiotemporal patterns, so they learn orthogonal universal bases offline, estimate basis weights online from sparse token samples, and add lightweight residual modules to recover texture and RoPE-related positional details. The method avoids forming the full attention matrix, preserves fidelity in complex motion, and reports strong speed-quality tradeoffs, including up to 1.92× speedup on HunyuanVideo and competitive results on Wan2.1 models while keeping visual quality close to baseline.

**Compliance With Llm Reviewing Policy:**

Affirmed.

**Final Justification:**

I have read the rebuttal and my concerns have been addressed. I will keep my score.

**Key Questions For Authors:**

**1. Can you compare your method with representative low-rank / linear attention baselines, not only sparse-attention methods?**
A convincing comparison would strengthen my assessment of the paper’s originality and empirical support.

**2. How universal are the learned bases across different model families, resolutions, and layers?**
If the bases transfer well beyond the reported settings, I would view the contribution as more significant and original.

**3. When does the sampling-based subspace projection fail, and are there practical ways to detect or mitigate such failures?**
A clear answer would improve my confidence in the method’s soundness and robustness, especially under OOD cases.

**Limitations:**

yes

**Strengths And Weaknesses:**

### Soundness

**Strength.**
The paper is empirically fairly strong: it evaluates on multiple large video diffusion backbones, reports both quality and efficiency metrics, includes component ablations, analyzes the speed–quality tradeoff for the number of bases, and provides additional robustness analysis for motion complexity and anchor sampling. Overall, the empirical evidence supports the claim that the method achieves a good efficiency–fidelity tradeoff.

### Presentation

**Weakness 1.**
The writing quality is uneven in places, with typos and awkward phrasing that reduce polish and occasionally distract from the technical content. This is fixable, but the paper would benefit from a careful proofreading pass.

### Significance

**Strength.**
The paper targets an important systems bottleneck in video diffusion transformers—quadratic attention cost—and shows practically meaningful improvements in latency and memory on large open models, including reduced memory footprints that move deployment closer to consumer hardware. This is a relevant and practically useful direction.

**Weakness 1.**
The demonstrated impact is currently concentrated on a specific setting—large VDiTs with this particular attention structure—so it is not yet clear how broadly the approach transfers across other architectures, resolutions, or video generation tasks.


### Originality

**Weakness 1.**
That said, several core ingredients—SVD-based low-rank approximation, projection from samples, and residual correction modules—are individually familiar. The novelty is therefore more in the particular integration and application to VDiTs than in introducing an entirely new primitive.

**Weakness 2.**
The claim of “universal bases” feels somewhat stronger than the evidence currently shown. The analysis is based on a limited set of backbones and uses resolution bucketing, which suggests the learned bases may still be somewhat model- or setting-dependent rather than fully universal.

**Weakness 3.**
The distinction from prior low-rank/linear attention methods is argued conceptually, but not isolated experimentally. Stronger originality claims would be easier to accept if the paper included direct comparisons or deeper analysis against those neighboring approaches, not only sparse-attention baselines.

---

> ### Author Rebuttal · Authors · 2026-03-31
>
> We sincerely thank the reviewer for the thoughtful and constructive feedback. We are encouraged that the empirical study and practical relevance were viewed positively. We agree that the paper can be strengthened in presentation, originality positioning, and robustness analysis, and we have prepared concrete revisions accordingly.
>
> (1) Presentation.
> We agree that the writing would benefit from a careful polishing pass. In the revision, we will thoroughly proofread the paper, fix typos and awkward phrasing, and sharpen several statements whose current wording is stronger than the evidence we intend to claim.
>
> (2) Comparison to representative low-rank / linear attention baselines.
> This is an excellent suggestion, and we agree that direct comparisons to neighboring low-rank/linear methods would substantially strengthen both originality and empirical positioning. To address this, we implemented representative zero-shot drop-in baselines on Wan2.1-1.3B (81 frames) under the same attention-replacement setting: Linformer, Nyströmformer, a SANA-inspired linear baseline, and our own “Basis-only” variant. The trend is clear and consistent: generic low-rank/linear approximations can obtain similar speedups, but they incur much larger fidelity degradation.
>
> Table 1: Comparison with representative low-rank / linear attention baselines.
>
> | Method | Attention Mechanism | Latency (s) $\downarrow$ | Speedup $\uparrow$ | PSNR $\uparrow$ | SSIM $\uparrow$ | LPIPS $\downarrow$ |
> |:---|:---|---:|---:|---:|---:|---:|
> | Original (Full) | Standard Softmax | 469 | 1.00x | 27.9 | 0.950 | 0.115 |
> | Linformer ($k=16$) | Static Low-Rank | 257 | 1.82x | 20.8 | 0.675 | 0.358 |
> | SANA-inspired | ReLU Linear Attn | 246 | 1.91x | 17.4 | 0.542 | 0.406 |
> | Nyströmformer | Landmark Approx. | 298 | 1.57x | 23.6 | 0.748 | 0.282 |
> | Ours (Basis only) | Plain SVD Basis | **240** | **1.95x** | 21.1 | 0.680 | 0.350 |
> | Ours (Full System) | Bases + Residual | 268 | 1.75x | **27.5** | **0.825** | **0.151** |
>
> This controlled comparison directly supports the main point of our paper: the contribution is not “using SVD” by itself, but making low-rank attention practically viable for high-fidelity video diffusion through the combination of offline data-driven bases, online projection, and structured Stage-3 residual correction.
>
> (3) Scope of “universal bases” and transferability.
> We agree that the phrase “universal bases” is stronger than the current evidence supports. In the revision, we will replace it with more precise wording such as “checkpoint-adaptive bases” or “shared structural motifs.” Our intended claim is not that one frozen numerical basis transfers unchanged across all models, resolutions, and settings; rather, it is that video attention exhibits reusable low-dimensional structure that can be efficiently captured and reused after lightweight calibration.
>
> To make this concrete, we add two analyses. First, within Wan2.1, adjacent layers show strong subspace overlap (typically >75%), which gradually decays with layer distance; this directly justifies our strided basis-sharing design. Second, in cross-basis reconstruction, bases extracted from a different checkpoint do produce higher error than matched bases, but still perform far better than a random orthogonal subspace of the same rank. Together, these results support a more accurate claim: VDiTs share nontrivial structural motifs, and Video-SVD exploits this shared structure while remaining checkpoint-adaptive where fidelity matters most. We will include these analyses and figures in the revision: [https://imgur.com/a/8nYcZqS].
>
> (4) When does subspace projection fail, and how can it be detected or mitigated?
> Our current paper already notes a failure mode on highly non-natural, high-frequency OOD inputs, where the low-rank approximation behaves like a smoothing filter.
>
> Beyond this observation, we now analyze the projection residual $r = \lVert D_{\text{sub}} \rVert_F / \left(\lVert A_{\text{sub}} \rVert_F + \varepsilon\right)$ as a runtime diagnostic. On challenging OOD cases, the hardest heads exhibit clearly elevated `r` compared with ID inputs. This enables a practical selective fallback mechanism: when a head’s residual exceeds a threshold, we route only that head back to exact FlashAttention. In a separate high-frequency OOD experiment, routing just 3% of the hardest heads improves PSNR from 17.5 dB to 23.9 dB (+6.4 dB) while preserving a 1.71x end-to-end speedup. This shows that the failure mode is not only detectable, but also systematically mitigable with a favorable systems tradeoff. We will add this analysis and discuss it explicitly in the revision and limitations section: [https://imgur.com/a/xL1XtmA].
>
> We believe these additions substantially strengthen the paper’s originality positioning, clarify the true scope of transferability, and make the robustness story much more complete. We thank the reviewer again for the valuable suggestions.

---

> > ### Author Rebuttal · Reviewer_oSbL · 2026-04-03
> >
> > I have read the rebuttal and my concerns have been addressed. I will keep my score.

---

### Official Review · Reviewer_HG1G · 2026-03-12

**Soundness:** 2
**Presentation:** 2
**Significance:** 2
**Originality:** 2
**Overall Recommendation:** 4
**Confidence:** 3

**Summary:**

This paper proposes Video-SVD, a method to accelerate Video Diffusion Transformers by approximating the attention computation using a set of learned basis patterns. The authors observe that attention maps in video models contain significant redundancy and can be represented using a small number of shared bases. They propose to use a dual-stream residual module to recover the lossed details. Experiments on HunyuanVideo and Wan2.1 show around 1.75–1.92× speedups while maintaining comparable generation quality with dense attention.

**Compliance With Llm Reviewing Policy:**

Affirmed.

**Final Justification:**

They addressed my concerns and questions successfully.

**Key Questions For Authors:**

See weaknesses.

**Limitations:**

Most baselines are not very new. The authors should compare with more state-of-the-art methods, such as SVG2 [1] or RadialAttention [2].
This will make the evaluation of the proposed method more solid.

[1] Sparse videogen2: Accelerate video generation with sparse attention via semantic-aware permutation

[2] Radial Attention: O(n\log n) Sparse Attention with Energy Decay for Long Video Generation

**Strengths And Weaknesses:**

Strengths:

1. The method can be applied to current models in a plug-and-play manner, without the need to modify their parameters.
2. Experiments show consistent improvements in speed and memory while maintaining similar visual quality.

Weaknesses:

1. The authors didn't explain why attention consistently exhibits a strong low-rank structure.
2. Offline preprocessing cost for learning the bases is not clearly discussed.
3. How well do the learned bases generalize to different datasets or video models?

---

> ### Author Rebuttal · Authors · 2026-03-31
>
> We sincerely thank the reviewer for the constructive feedback and helpful suggestions. We are encouraged that the plug-and-play nature of our method and its efficiency gains were recognized. Below we address the four concerns and will incorporate all clarifications in the revision.
>
> (1) Why does attention exhibit a strong low-rank structure?
> We agree that the original paper explained the empirical low-rank phenomenon more clearly than its intuition. Our claim is an **effective** low-rank structure rather than exact rank deficiency. In video, many tokens correspond to a small number of shared semantic entities (e.g., background regions, object parts, motion trajectories), which makes many rows/columns of the pre-softmax matrix highly correlated, up to high-frequency residuals such as RoPE textures.
>
> This interpretation is consistent with the rapid singular-value decay shown in Fig.2 in the paper. To make the intuition more concrete, we will add an additional reordered pre-softmax matrix visualization on Wan2.1 [https://imgur.com/a/WrtFHyo] showing that, after grouping similar tokens, 56.2% of the interaction energy concentrates into only 18.1% of the matrix area (3.1x density). This supports the view that dense attention spends much of its computation on redundant interactions among semantically similar token groups.
>
> (2) Offline preprocessing cost for basis learning.
> We agree the offline preprocessing cost was underspecified in the submission. This step is a one-time, checkpoint-and-resolution-bucket-specific calibration based on parameter-free streaming Incremental SVD, after which the learned bases can be reused for all subsequent inference. Due to the character limitation here, we will report the calibration budget clearly in the revision, including i) the number of sampled matrices and sampling protocol, ii) one-time collection and SVD time, and iii) compressed basis storage. As an example, on Wan2.1-1.3B,  the basis bank (including 10800 matrices) is collected across videos, layers, timesteps, motion tiers, and heads costing ~18 GPU hours, and compressed ~17.2 TB matrices into ~23 GB of reusable bases, costing 75 CPU hours.
>
> (3) Generalization across datasets and video models.
> We agree that the phrase “universal bases” in the draft may **incorrectly suggest** direct cross-model basis transfer. This is not our claim, because different VDiTs consist of different hidden dimensions, head counts, and feature spaces. Saying “universal”, we mean “universal bases” across the whole model including layers, heads, timesteps and etc. Our claim is **method-level generalization**: the same pipeline (offline basis extraction + online projection + residual compensation) works across distinct backbone families, as shown on both Wan2.1 and HunyuanVideo. For a new checkpoint, we perform a one-time calibration, i.e., basis extraction and, if used, lightweight residual-module fitting.
>
> To address dataset transfer more directly, we additionally evaluated zero-shot transfer by using bases/MLPs extracted only from VBench and applying them to unseen datasets with clear domain shift:
>
> | Dataset | Full-Attn PSNR | Video-SVD PSNR | Drop |
> |:---|:---:|:---:|:---:|
> | UCF-101 | 28.52 | 28.15 | -0.37 |
> | DEVIL | 27.24 | 26.71 | -0.53 |
> | EvalCrafter | 26.85 | 26.38 | -0.47 |
>
> The degradation is modest, without structural collapse or severe artifacts. We will therefore replace “universal bases” with a more precise description such as “**shared low-dimensional structural motifs**” and “**checkpoint-adaptive calibration**”. At the same time, we will state the OOD limitations such as highly non-natural, high-frequency domains more explicitly.
>
> (4) Comparison with newer baselines such as SVG2 and Radial Attention.
> We completely agree that these methods should be included, and we will add both SVG2 and Radial Attention as formal baselines in the revised paper. At the same time, we would like to clarify that Video-SVD is orthogonal to these methods in both mechanism and deployment. Radial Attention relies on a predefined geometric decay prior, and SVG2 relies on semantic token permutation/clustering with custom sparse execution. In contrast, Video-SVD reconstructs attention through data-driven global bases, online coefficient solving, and structured residual compensation. It uses only standard PyTorch operations, does not require custom CUDA kernels, and remains training-free at inference time. Due to the limited rebuttal window, we prefer not to draw rushed head-to-head conclusions before completing strictly controlled experiments under matched hardware settings to avoid over-claiming (their original results are provided in the anonymous link: [https://imgur.com/a/3X4NX9o]). In the revision, we will add these baselines to the main tables and explicitly discuss their conceptual and engineering differences.
>
> We hope these clarifications address the reviewer’s concerns, and we thank the reviewer again for helping us improve the paper.

---

> > ### Author Rebuttal · Reviewer_HG1G · 2026-04-03
> >
> > I have raised the score accordingly.

---

### Official Review · Reviewer_5frv · 2026-03-13

**Soundness:** 3
**Presentation:** 2
**Significance:** 2
**Originality:** 3
**Overall Recommendation:** 4
**Confidence:** 5

**Summary:**

This paper proposes Video-SVD, an attention acceleration method for video diffusion transformers. The core idea is to replace rigid spatial/temporal sparse priors with a global low-rank basis reconstruction of attention, followed by a dual-stream residual compensation module for content and positional details. The method is evaluated on HunyuanVideo and Wan2.1 (1.3B / 14B), showing consistent speedups while preserving fidelity better than sparse baselines such as STA, PAB, VORTA, and SVG.

**Compliance With Llm Reviewing Policy:**

Affirmed.

**Final Justification:**

I have read the rebuttal and my concerns have been addressed. I will keep my score.

**Key Questions For Authors:**

The paper mentions using prompts from Penguin Benchmark with VBench prompt optimization, but the exact number of evaluation prompts and videos is not clearly specified in the main paper.

**Limitations:**

Yes.

**Strengths And Weaknesses:**

Strengths:

The paper addresses an important practical problem, namely efficient attention for large-scale video diffusion transformers. The motivation is clear: many video attention heads exhibit hybrid spatio-temporal patterns, which are not well captured by rigid spatial-vs-temporal sparsity assumptions.

The proposed approximation is fairly well motivated. Modeling attention through a small number of global low-rank bases, followed by lightweight residual compensation, is a reasonable design choice and seems better aligned with complex motion patterns than hard sparse masking.

The ablation study shows that the residual compensation modules are essential for recovering quality. Experimental results on multiple modern VDiT backbones show a favorable quality-efficiency trade-off.

Weaknesses:

My main concern is the generalization ability of the proposed method. The framework depends on both offline extracted bases and learned MLP-based residual modules, and both components may be sensitive to distribution shift. If test-time attention patterns differ significantly from those covered during offline basis extraction, or if the residual correction modules encounter cases outside their training regime, the approximation may break down. In this sense, the method may be less reliable on genuinely unseen cases, and the current paper does not yet provide sufficient evidence on such out-of-distribution robustness.

The paper demonstrates promising results on the evaluated backbone families, but does not provide enough evidence on transfer across broader architectures, resolutions, or more clearly shifted video domains.

The paper states that the bases are extracted from a “large-scale” collection of attention matrices sampled from pre-trained VDiT models, but it does not clearly report the actual scale of this basis-construction dataset, such as how many prompts, videos, or attention matrices were used, nor how they were sampled across layers, heads, or timesteps. This makes it harder to assess the robustness and reproducibility of the offline basis extraction process.

---

> ### Author Rebuttal · Authors · 2026-03-31
>
> We thank the reviewer for recognizing the motivation and design of our method. We agree that generalization, OOD robustness, and reproducibility are central to its reliability, and we will clarify these points in the revision.
>
> (1) Generalization under distribution shift.
> To directly address the concern on distribution shift, we conducted additional zero-shot transfer experiments: the offline bases and residual MLPs were extracted/trained only on VBench, and then applied directly to unseen datasets with clear domain shift.
>
> | Dataset | Full-Attn PSNR | Video-SVD PSNR | Drop |
> |---|---:|---:|---:|
> | UCF-101 | 28.52 | 28.15 | -0.37 |
> | DEVIL | 27.24 | 26.71 | -0.53 |
> | EvalCrafter | 26.85 | 26.38 | -0.47 |
>
> The degradation is modest (~0.46 dB on average), without structural collapse or severe artifacts. We will therefore add these results to the revision and replace “universal bases” with a more precise description, such as “shared low-dimensional structural motifs” and “checkpoint-adaptive calibration”. At the same time, we will state the OOD limitations—such as clearly shifted high-frequency synthetic domains—more explicitly.
>
> (2) Transfer across models and resolutions.
> We will also clarify an important distinction: we do  not claim direct cross-model basis transfer, because different VDiTs have different hidden dimensions, head counts, and feature spaces. Our claim is **method-level generalization**: the same pipeline (offline basis extraction + online projection + residual compensation) works across distinct backbone families, as already demonstrated on both Wan2.1 and HunyuanVideo. Each new checkpoint requires only a lightweight one-time checkpoint-adaptive calibration, i.e., basis extraction and, if used, lightweight residual-module fitting.
>
> For resolution, our method is designed with resolution bucketing. We observe robustness under moderate token-count shifts within neighboring resolution buckets, because the dynamic subspace projection can absorb moderate structural variation. However, large structural shifts across distant buckets eventually degrade fidelity, since the learned bases are no longer well aligned with the new token geometry. We will make this boundary explicit in the revision, discuss it as a limitation rather than over-claiming broad resolution invariance, and add a small cross-bucket transfer analysis to quantify this boundary more clearly.
>
> (3) Offline preprocessing cost and calibration protocol.
> Thank you for pointing out that the offline basis-learning cost was not clearly discussed. This preprocessing is a strictly one-time, amortized cost using parameter-free streaming Incremental SVD. To balance robustness and storage, we curate a global pool of 600 VBench videos, from which we allocate a stratified subset to each layer across 5 motion tiers. For each allocated video, we sample 3 denoising steps (early/mid/late) and exactly 2 attention heads (via round-robin to globally saturate head coverage per layer). This yields 10,800 matrices for Wan2.1-1.3B (30 layers x a subset of 60 videos x 3 timesteps x 2 heads) and 19,200 matrices for Wan2.1-14B (40 layers x a subset of 80 videos x 3 timesteps x 2 heads).
>
> For Wan2.1-1.3B, collection takes ~18h on 8xA800 and streaming SVD takes ~75h on CPU, for ~93h total; compressed bases require ~23 GB instead of ~17.2 TB dense storage. For Wan2.1-14B, collection takes ~38h and SVD ~130h, for ~168h total; compressed bases require ~45 GB instead of ~30.7 TB. For HunyuanVideo, the pipeline scales predictably to ~7.5 days while peak RAM remains bounded by mini-batch streaming. We will add this budget analysis in the revision.
>
> (4) Exact evaluation benchmark details.
> Thank you for pointing out this omission. We will make the evaluation setup explicit in the main paper. Specifically, we use exactly 600 text-to-video prompts from the Penguin Benchmark, after prompt optimization by VBench, and generate exactly 600 videos for each evaluated model architecture.
>
> These prompts cover: subject dynamics (complex animal behaviors and detailed human actions), spatial relationships (precise relative positioning of multiple objects), camera movements (medium-long shots, wide-angle shots, explicit panning), and visual styles (4K realistic videography, 3D cartoon, sci-fi art, and anime). We will document both the exact count and this semantic coverage in the revised Experimental Setup. To avoid confusion, this 600-prompt evaluation set is separate from the 600-video calibration pool used for basis extraction.
>
> We believe these clarifications make the scope, robustness, and reproducibility of Video-SVD much clearer. We thank the reviewer again for the helpful suggestions.

---

> > ### Author Rebuttal · Reviewer_5frv · 2026-04-02
> >
> > I have read the rebuttal and my concerns have been addressed. I will keep my score.

---

### Decision · Program_Chairs · 2026-04-30

**Decision:**

Accept (regular)

**Comment:**

This paper presents Video-SVD that overcomes the quadratic computational bottleneck of self-attention by replacing rigid spatial/temporal priors with a global, low-rank basis reconstruction. Observing that dense video attention patterns inherently reside on a low-rank manifold and exhibit hybrid spatiotemporal interactions , the method extracts universal orthogonal bases offline using SVD.

Reviewers result in scores of 4, 4, 4, and 5; appreciated significant e2e speedups and reduces memory footprints sufficiently. While initial concerns were raised regarding the "universality" of the bases and the lack of comparisons to standard linear attention baselines , the authors provided a extensive rebuttal that clarified the concerns.

Overall, AC also sees this work’s contribution as solid and agrees with its acceptance.